

# Methane oxidation potential of soils in a rubber plantation in Thailand affected by fertilization

Jun Murase[1,2], Kannika Sajjaphan[2], Chatprawee Dechjiraratthanasiri[2], Ornuma Duangngam[3], Rawiwan Chotiphan[4], Wutthida Rattanapichai[2], Wakana Azuma[5], Makoto Shibata[6], Poonpipope Kasemsap[7], Daniel Epron[7,8]

[1]Graduate School of Bioagricultural Sciences, Nagoya University, Nagoya, 464-8601, Japan

[2] Department of Soil Science, Faculty of Agriculture, Kasetsart University, Bangkok, 10900, Thailand

[3] Center of Thai-French Cooperation on Higher Education and Research (DORAS Center), Kasetsart University, Bangkok 10900, Thailand

[4] Sithiporn Kridakara Research Station, Faculty of Agriculture at Kamphaeng Saen, Kasetsart University, Prachuap Khiri khan 77170, Thailand

[5]Graduate School of Agricultural Science, Kobe University, Kobe, 657-8501, Japan

[6]Graduate School of Global Environmental Studies, Kyoto University, Kyoto, 606-8501, Japan

[7] Department of Horticulture, Faculty of Agriculture, Kasetsart University, Bangkok, 10900, Thailand

[8]Graduate School of Agricultural Science, Kyoto University, Kyoto, 606-8501, Japan

*Correspondence to*: Jun Murase (murase@agr.nagoya-u.ac.jp) and Kannika Sajjaphan (agrkks@ku.ac.th)

**Abstract.** Forest soils, as crucial sinks for atmospheric methane in terrestrial ecosystems, are significantly impacted by changes in ecosystem dynamics due to deforestation and agricultural practices. This study investigated the methane oxidation potential of rubber plantation soils in Thailand, focusing on the effect of fertilization. The methane oxidation activity of the top soils (0-10 cm) in the dry season was found to be extremely low and slightly increased in the wet season, with lower activity for higher fertilization levels. The potential methane oxidation potential of the topsoil was too low to explain the in-situ methane uptake. Soils below 10 cm depth in unfertilized rubber plantations showed higher activity than the surface soils, and methane oxidation was detected at least down to 60 cm depth. In contrast, soils under the high-fertilization treatment exhibited similarly low activity of methane oxidation up to 60 cm depth as surface soils both in dry and wet seasons, indicating that fertilization of para rubber plantation negatively impacts the methane oxidation potential of the soils over the deep profile without recovery in the off-season with no fertilization. Methane uptake per area estimated by integrating the methane oxidation potentials of soil layers was comparable to the field flux data, suggesting that methane oxidation in the soil predominantly occurs in depths below the surface layer. These findings have significant implications for understanding the environmental impacts of tropical forest land uses on methane dynamics and underscore the importance of understanding methane oxidation processes in soils.





## 1 Introduction

Methane is the most important anthropogenically enhanced greenhouse gas in the atmosphere after $CO_2$ (Forster et al., 2021).
It is thus important to fully quantify and characterize all sources and sinks to include the role of terrestrial ecosystems in
mediating atmospheric exchange. Unsaturated aerobic soils are important sinks of atmospheric methane via oxidation by
methane-oxidizing bacteria with a global estimation of 11–49 Tg $CH_4$ $yr^{-1}$ (Saunois et al., 2020). The global mean methane
uptake rate in forest soils is reported to be $3.95 \pm 1.78$ kg $CH_4$ $ha^{-1}$ $yr^{-1}$, with a total sink of $14.98 \pm 6.75$ Tg $CH_4$ $yr^{-1}$ in 1999–
2020, thus playing an essential role in the terrestrial methane sink (Feng et al., 2023). Temperate and tropical forest soils are
the predominant sinks, contributing 84% of total methane sink in forest ecosystems (Feng et al., 2023).

Forest conversion is suspected of weakening the sink of atmospheric methane (Verchot et al., 2000). Rubber plantations in
tropical regions have been expanding worldwide, particularly in Asia, where lowered ecosystem functions compared to forests
have been demonstrated (Singh et al., 2021). Deforestation and following agricultural use of a tropical forest, such as para
rubber and oil palm plantations, tend to decrease the methane sink of soils (Lang et al., 2019; Lang et al., 2020; Aini et al.,
2020; Lang et al., 2017; Zhou et al., 2021). The large-scale expansion of rubber plantations in Southeast Asia has decreased
methane uptake by soil. However, a mechanistic understanding of the associated processes within the soil profile is still missing
(Lang et al., 2020).

Monitoring the surface methane flux has been used to study the methane uptake of soils in tropical forests and the conversion
effects of land use on it. Rubber monocultures showed lower rates of methane uptake than natural forests (Werner et al., 2006),
which could turn the rubber soils into methane emitters during a certain period of the wet season (Lang et al., 2019). The water-
50 filled pore space, which is increased by rubber plantation and with rubber age, is correlated with the methane flux, suggesting
that soil compaction by agricultural machinery may suppress methanotrophy and promote methanogenesis by reducing gas
exchange (Lang et al., 2019). Methane uptake rates in the tropical forests in Indonesia under deforestation and rubber plantation
are negatively affected by clay content that affects the soil pore space (Ishizuka et al., 2002).

Mineral nitrogen (ammonia and nitrate) is one of the most critical factors controlling aerobic methane oxidation in soil
(Bodelier and Laanbroek, 2004; Bodelier, 2011). Meta-analyses demonstrate that nitrogen enrichment increases methane
emissions in most soil ecosystems, and nitrogen fertilization significantly reduces methane uptake (Liu and Greaver, 2009;
Aronson and Helliker, 2010). However, the effect of nitrogen on methane uptake by soil can vary depending on different
factors, such as the enrichment level of nitrogen, ecosystem, biome, and duration of fertilization (Aronson and Helliker, 2010).
Low rates of nitrogen addition in forest and tree plantation systems occasionally stimulate methane oxidation (Geng et al.,
2017; Koehler et al., 2012); the first study to demonstrate the role of soil fertility on methane uptake in a tropical landscape
reported nitrogen limitation of methane uptake in the converted land-use types including rubber plantation (Hassler et al.,
2015). Another study demonstrated no significant contribution of ammonium nitrogen to predicting methane flux (Lang et al.,
2019). Determining the effects of fertilizer application in rubber plantations on soil methane oxidation processes is essential





to exploring soil management to compromise between natural rubber production and the maintenance of soil ecosystem
function as a methane sink.

In this study, we measured the potential rates of soil methane oxidation using a microcosm incubation experiment to understand the effect of land use change and fertilization management on soil methane uptake in a para rubber plantation. While most studies assume that methane oxidation in forest soils occurs primarily in the surface soil, we also targeted the deeper soil layers because our initial experiment indicated that the surface soils had only slight activity of methane oxidation when compared to the in-situ methane fluxes. Our results suggest that methane oxidation in the deep soils is considerably involved in the methane dynamics in the rubber plantation and is negatively influenced by fertilization to the surface soil.

## 2 Materials and Methods

### 2.1 Study site

The study mainly targeted a para rubber plantation site in Sithiporn Kridakorn Research Station, Kasetsart University (SKRS, N 10°59.3', E 99°29.3'), Prachuap Khirikhan province, located in southern Thailand. The climate of the region is tropical monsoon with a mean annual precipitation of 1700 mm (2010-2023), falling mainly in the hot and wet season from April to September. The rainiest months are October and November, with more than 250 mm on average. The driest month is January, which is 64 mm on average. The main features of the soil in the SKRS were deep soil with a sandy-loam texture, low water retention capacity, poor organic matter content, and low cation exchange capacity due to the dune origin. The soil is classified as Rhodic Kandiudults.

The rubber plantation of the SKRS site has four different levels of fertilizer treatment with randomized four replicate blocks (A–D): Tr1, no; Tr2, low; Tr3, intermediate; Tr4, high (Table 1). Tr3 falls within the range of the recommended fertilizer application rates for mature rubber plantations in Thailand by Thai public institutions; recommendations exceeded by 40% of rubber farmers (Chambon et al., 2018). Chemical fertilizers of nitrogen, phosphorus, and potassium are top-dressed in the wet

**Table 1** Sampling sites of the Sithiporn Kridakorn Research Station

| Vegetation | Treatment | Site replication | Fertilization (N/P/K) | | Soil sampling depth | | |
| --- | --- | --- | --- | --- | --- | --- | --- |
| | | | Early rainy season | Late rainy season | Feb 2023 (dry season) | Aug 2023 (rainy season) | Feb 2024 (dry season) |
| Para rubber | T1 | 4 (Blocks A-D) | 0 | 0 | 0-10 cm (Block B) | 0-10 cm (Blocks A-D) 0-55 cm (Block B, 8 layers) | 0-50 cm (Blocks A-D, 5 layers) |
| | T2 | 4 (Blocks A-D) | 500g/tree 15/9/20 | 0 | 0-10 cm (Block B) | 0-10cm (Blocks A-D) | 0-50 cm (Blocks A-D, 5 layers) |
| | T3 | 4 (Blocks A-D) | 500g/tree 21/7/14 | 500g/tree 15/9/20 | 0-10 cm (Block B) | 0-10cm (Blocks A-D) | 0-50 cm (Blocks A-D, 5 layers) |
| | T4 | 4 (Blocks A-D) | 850g/tree 21/7/14 | 850g/tree 15/9/20 | 0-10 cm (Block B) | 0-10cm (Blocks A-D) 0-60cm (Block B, 5 layers) | 0-50 cm (Blocks A-D, 5 layers) |
| Palm | Litter | 3 | NA | NA | - | 0-10 cm | - |
| | No Litter | 3 | NA | NA | - | 0-10 cm | - |
| Forest | - | 4 | 0 | 0 | - | 0-10 cm | - |

season, evenly to half of the area between the planting rows. A secondary forest and oil palm plantation adjoining the rubber



plantation in SKRS were also studied for comparison. The bulk density of the soils ranged from 1. 4 g cm$^{-3}$ (forest and palm) to 1.5 g cm$^{-3}$ (para rubber); the surface of the palm soil covered with litter had a low density (0.94 g cm$^{-3}$).

A rubber plantation in Chachoengsao Rubber Research Center (CRRC, N 13°33.9', E101°27.3'), Chachoengsao province, located in central Thailand, was also studied to compare the effect of fertilization on soil methane oxidation with SKRS. The annual precipitation is 1400 mm on average for 2022-2023. The soil is classified as clayey-skeletal, kaolinitic, isohyperthermic Typic Kandiustults. Some physicochemical properties of the CRRC soils have been reported before (Kanpanon et al., 2015; Satakhun et al., 2013). The plantation received chemical fertilizer twice yearly at 500 g per tree (N: P: K = 30: 5: 18) in the middle of the inter-row.

## 2.2 Sample collection

Soil samples were collected from 0-10 cm layers in the middle of the inter-row of the rubber plantation in February 2023 and 2024 (dry season) and August 2023 (wet season). In February 2023, triplicate soil samples were collected from one of the replicate blocks (block B). In August 2023 and February 2024, a soil sample was collected from each block, giving four replicate samples per treatment. Soil samples with up to 60 cm depths were also collected at the SKRS rubber plantation site: from Tr1 and Tr4 of block B in August 2023 and all treatments of all replicate blocks in February 2024. Topsoils (0-10 cm) in the forest and palm plantation in SKRS and CRRC rubber plantation were collected in August 2023. Four replicate sites of the forest were randomly selected, and in the palm plantation, the two contrasting locations, i.e., with and without litter cover, were selected in triplicate. In CRRC, no fertilization experiment was conducted; thus, we selected locations in the middle of the inter-row for fertilized soils and those in the middle of the row for unfertilized soils. The soil samples were sieved (< 2 mm) to measure methane oxidation potential.

## 2.3 Methane oxidation potential

The potential methane oxidation rates (PMORs) of the soils were determined by a microcosm incubation experiment. Ten grams of sieved soils were put into 50-ml or 100-ml GC vials (Nichiden-Rika Grass, Kobe, Japan). The vials were capped with butyl rubber stoppers and open-top screw caps and injected with 0.5 or 1.0 ml of 1% methane to give an initial concentration of 50 ppmv in the headspace containing atmospheric air. The samples were incubated in the dark at 25°C. Gas samples (0.25 ml) were periodically sampled from the headspace, and methane concentrations were measured with a gas chromatograph with a flame ionization detector (GC-2014, Shimadzu, Kyoto, Japan). The PMORs were calculated from the linear regression of the methane concentration decreasing with incubation time. The potential methane uptake rate per area was estimated by adding up the methane oxidation rates of different layers, assuming the bulk density of the soil *in situ* as 1.5 g cm$^{-3}$ according to our pilot survey.





### 2.4 Soil methane flux

Soil methane fluxes of the SKRS rubber plantation were measured a few days before or after soil sampling on 24 (6 replicates ×4 blocks) PVC collars (20 cm in diameter and 13 cm in height) in each fertilization treatment Collars were covered with a soil chamber (Li 8100-103, Li-Cor; Lincoln, USA). Methane flux was calculated from the rate of change in the methane concentration measured using a trace gas analyzer (Li 7810, Li-Cor) as described in (Epron et al., 2023).

### 2.5 Soil chemical analysis

Soil pH was measured in distilled water (soil-water ratio of 1:1). Soil electrical conductivity was measured using an EC meter after missing soil with fivefold distilled water. Soil organic carbon was measured using the wet oxidation method (Walkley and Black, 1934), and soil total nitrogen was measured using the Kieldahl method (Bremner, 1996).

### 2.6 Statistics

Linear regression analysis was performed using Origin 2024 (OriginLab Corporation, Northampton, USA). Differences in PMORs and soil methane fluxes between treatments were tested using a one-way variance analysis and the Kruskal-Wallis test with the post hoc Tukey's HSD and Dunn-Bonferroni tests, respectively (SPSS for Windows, version 22.0).

## 3 Results and discussion

### 3.1 Methane oxidation potential of the surface soils

We first focused on the methane oxidation potentials of the 0-10 cm layer of soil, based on previous findings that atmospheric methane uptake in soils is highest in the surface layer (Lang et al., 2020). The soil samples in the dry season (February 2023) showed minor methane oxidation; only a slight decrease in methane concentration was observed in the site with no fertilization

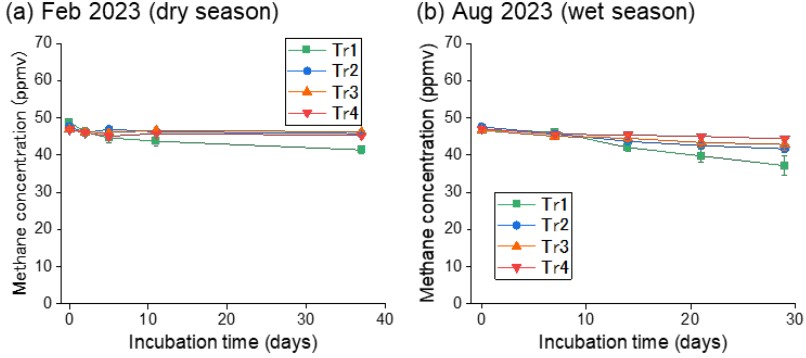

**Figure 1:** Methane oxidation of surface (0-10 cm) soils of the para rubber plantation in (a) February and (b) August 2023. The February data represent the average and standard error (n=3) of block B, and the August data represent the average and standard error (n=4) of the four blocks (blocks A to D).



(Tr1, Figure 1a). In the wet season (August 2023), a slight but detectable decrease in methane with time was observed in all treatments compared to the dry season, and Tr1 showed a more significant decrease in methane than the other treatments (Figure 1b).

The soils collected in August 2023 from sites other than the SKRS rubber plantation showed methane oxidation rates with larger variations (Supplementary Figure S1). The forest soils demonstrated a significant spatial variation in methane oxidation. Palm soils showed linear methane consumption, and no significant effect of the litter cover was observed. Methane consumption of CRRC rubber plantation soils sampled in the planting rows was higher than those sampled in the middle of the inter-rows where fertilizer is spread.

The calculated PMORs for the topsoils of August 2023 ranged between 0.02 and 1.23 ngCH$_4$ g$^{-1}$ dry soil hr$^{-1}$ (Figure 2). The PMORs of para rubber soils in SKRS ranged from 0.02 to 0.17 ngCH$_4$ g$^{-1}$ dry soil hr$^{-1}$ with higher rates in Tr1 (no fertilization) than Tr4 (highest fertilization) ($p < 0.05$), suggesting that fertilization with a critical level gives a negative impact on soil

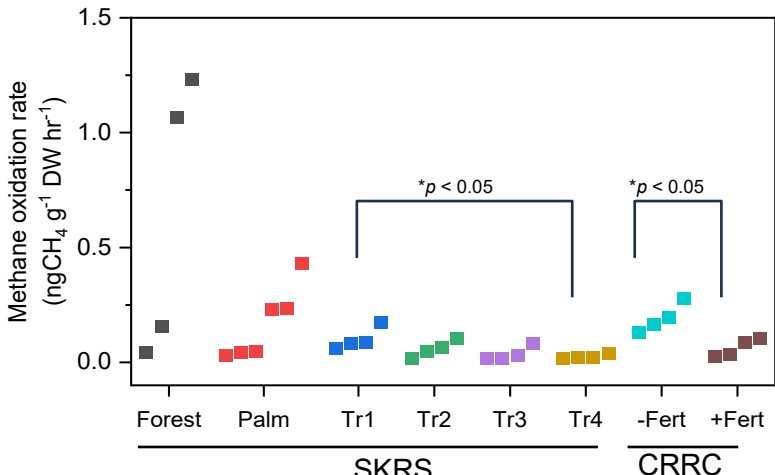

**Figure 2:** Potential methane oxidation rate of the surface (0-10 cm) soils corrected in August 2023 (wet season). The unfertilized soils (Tr1 in SKRS and –Fert in CRRC) have higher rates than the fertilized soils (Tr4 and +Fert, respectively).

methane oxidation, which is consistent with previous studies of forest soils (Liu et al., 2024; Aronson and Helliker, 2010). The PMORs of the SKRS rubber soils are among the lowest values for European forest soils (Täumer et al., 2021). The rubber soils in CRRC from no fertilized sites (planting rows) also showed higher methane oxidation rates than those from the fertilized sites (inter-rows, $p < 0.05$). The forest soils showed the most considerable spatial variation; two of the four recorded the highest PMORs in this study, which were in the higher range of European forest soils (Täumer et al., 2021). The palm soils also showed considerable variation, and litter cover had no effect. The significant differences in PMORs between different land uses in the SKRS were not detected due to the large spatial variation in the forest and palm soils.

There was no clear relationship between the PMORs and soil chemical parameters measured in this study. However, a weak positive correlation with total nitrogen was detected (Supplementary Figure S2), which may imply that soil organic nitrogen





slowly supplies inorganic nitrogen that does not suppress but sustains methane-oxidizing bacteria (Geng et al., 2017) in contrast
to the high application of chemical fertilizers.

The in situ methane flux in the SKRS rubber plantation soil was negative in the dry season (February 2023 and 2024),
indicating that methane oxidation predominates methane production, and the soil functioned as a net methane sink. Middle
and high fertilization (Tr3 and Tr4) suppressed the in-situ soil methane uptake compared to no fertilization (Tr1, Table 2),
consistent with the results of PMORs. In the wet season (August 2023), the methane fluxes in Tr1 and Tr2 were comparable

to those in the dry season, while Tr3 and Tr4 showed positive methane fluxes on average, indicating that methane production
in the soil exceeds methane oxidation as reported before (Lang et al., 2019). The estimated aerial PMORs of the surface soil
(0-10 cm) were much lower than the methane flux on site in the dry season; the same trend was observed in Tr1 and Tr2 in the
wet season.  The result suggests a significant gap between the PMOR of topsoils and the methane uptake *in situ*.

Table 2 In situ methane flux and PMOR of surface soil (0-10 cm) in SKRS

|  | Treatment | Methane flux ($nmol\ m^{-2}\ s^{-1}$) | n | PMOR ($nmol\ m^{-2}\ s^{-1}$) | n |
|---|---|---|---|---|---|
| Feb 2023 | Tr1 | -0.503 ± 0.029 [a] | 24 | 0.145 ± 0.014 [a] | 3 |
| (dry season) | Tr2 | -0.479 ± 0.044 [ab] | 24 | 0.039 ± 0.002 [b] | 3 |
|  | Tr3 | -0.338 ± 0.046 [b] | 24 | 0.015 ± 0.008 [b] | 3 |
|  | Tr4 | -0.121 ± 0.065 [c] | 24 | 0.031 ± 0.001 [b] | 3 |
| Aug 2023 | Tr1 | -0.562 ± 0.090 [a] | 24 | 0.263 ± 0.065 [a] | 4 |
| (wet season) | Tr2 | -0.522 ± 0.074 [a] | 24 | 0.156 ± 0.046 [ab] | 4 |
|  | Tr3 | 0.122 ± 0.281 [b] | 24 | 0.096 ± 0.040 [ab] | 4 |
|  | Tr4 | 0.058 ± 0.151 [b] | 24 | 0.063 ± 0.013 [b] | 4 |
| Feb 2023 | Tr1 | -0.764 ± 0.037 [a] | 24 | 0.037 ± 0.007 [a] | 4 |
| (dry season) | Tr2 | -0.542 ± 0.117 [a] | 24 | 0.029 ± 0.006 [ab] | 4 |
|  | Tr3 | -0.393 ± 0.071 [b] | 24 | 0.013 ± 0.002 [b] | 4 |
|  | Tr4 | -0.216 ± 0.051 [b] | 24 | 0.015 ± 0.005 [ab] | 4 |

The values are means±standard errors. Values of different letters are significantly
different among the treatments in respective seasons ($P$ <0.05)

### 3.2 Methane oxidation potential of the soils collected from deep layers

Then, we tested our hypothesis that the deeper soils could contribute to methane consumption. The soils collected from the
layers deeper than 10 cm in the rubber plantation in the dry season (February 2024) showed active methane oxidation compared
to the topsoil (0-10 cm) in Tr1 (no fertilization) (Figure 3 and Supplementary Figure S3); one exception was Block D where
the methane consumption of the topsoil was comparable to the deeper layer soils (Figure S3). Active methane consumption in
the deeper soils was also observed in Tr 2 except for one replicate block (Block A, Figure. S3). Methane oxidation was less

active throughout the soil profiles in Tr3 and Tr4 than in Tr1 and Tr2, except for Tr3 in Block B, which showed more methane
oxidation in the deeper layer (10-20 cm, Figure 3). The active methane oxidation in the deeper layers of Tr1 and low methane
oxidation throughout the soil profiles in Tr4 were also observed in the wet season (Supplementary Figure S4).



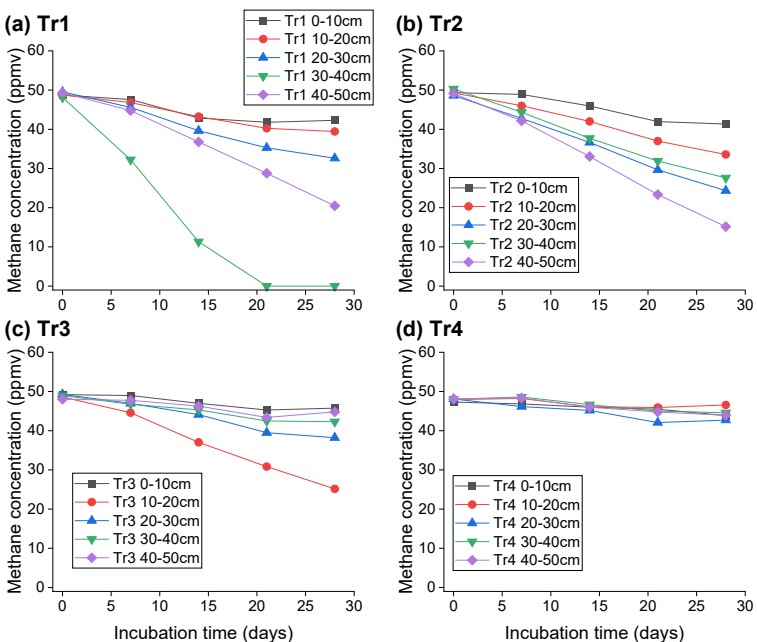

**Figure 3:** Methane oxidation of soils corrected from different depths in the rubber plantation (block B) in February 2024 (dry season).

We measured PMORs for 22 samples of deeper layers (> 10 cm) from Tr1. Among them, 20 samples showed higher PMORs than the topsoils (up to 10 cm) of the same site and sampling time (Figure 4 and Supplementary Figure S5), up to 30 times

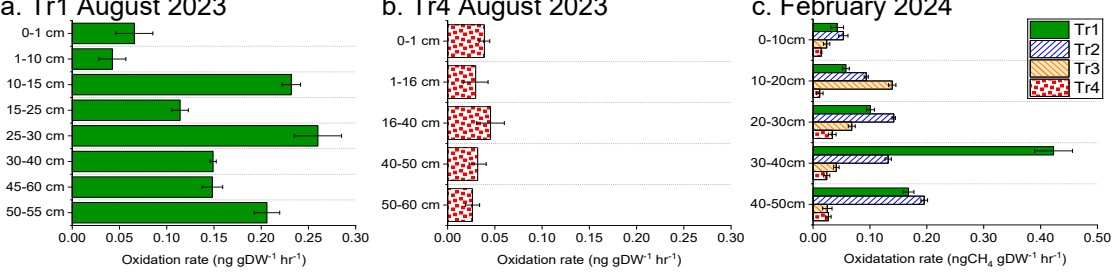

**Figure 4:** Depth profile of potential methane oxidation rate of the SKRS rubber plantation soils (Block B). The error bars are based on the error of the regression slope. August is in wet season and February in dry season.

higher. On the other hand, only three samples of deeper layers from Tr4, out of 20 samples in total, showed slightly higher PMORs than the top soils, and the rest of the samples showed PMORs comparable to or less than the topsoils. Tr2 and Tr3 had 75 % (12 of 16) and 50 % (8 of 16) of the deeper layers that showed higher PMORs than the topsoils.

The estimated potential methane uptake rates per area of the SKRS rubber plantation soil tended to decrease along the fertilization level (Figure 5), and Tr1 and Tr2 showed higher methane uptake rates than Tr3 and Tr4 ($p < 0.05$). The estimated





rates ranged between 0.24 (Tr4) and 2.21 (Tr1) nmol m$^{-2}$ s$^{-1}$, which exceeded the in-situ fluxes that represent net methane uptake, i.e., the balance between oxidation and production (Table 2). Our results demonstrated that 1) unlike the previous studies, the deeper layer soils in the rubber plantation would contribute to methane oxidation, and 2) fertilization suppresses the soil methane oxidation potentials throughout the soil profile at least up to 60 cm.

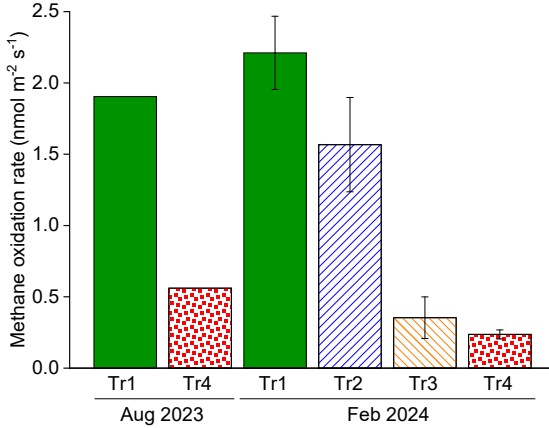

**Figure 5:** Estimated areal methane oxidation potential of SKRS rubber plantation under different fertilization. Data in August 2023 (wet season) were obtained from block B, and those in February 2024 (dry season) were the average and standard error of the four replicate blocks (blocks A to D).

In the SKRS rubber station, Tr3 and Tr4 received more fertilizer than Tr2. In addition, Tr2 received fertilizer only once a year at the beginning of the wet season, while Tr3 and Tr4 received fertilizer again in the late wet season. Fertilization, especially nitrogen fertilizer application, is often reported to inhibit soil methane oxidation (Täumer et al., 2021; Bodelier, 2011; Bodelier and Laanbroek, 2004). In addition to the high amount of fertilization, recurring and prolonged disturbances of methane oxidation by fertilization in Tr3 and Tr4 may outcompete the resilience of methane oxidation (Lim et al., 2024). Notably, fertilizers applied on the surface had a suppressive effect on methane oxidation in the deeper layers, at least up to 60 cm.

## 4 Conclusion

Our results provide a new insight into the impact of agricultural land use of tropical forests on the ecological function in a greenhouse gas cycle. Even a recommended fertilizer application rate in a rubber plantation could hurt soil methane oxidation potential spatially and temporarily vastly, which may change the methane cycle of tropical forests. The harmonized land use of tropical forests for rubber plantation in the future should include the risk of reduced methane uptake due to fertilization. As soil organic nitrogen had a weak but positive correlation with soil methane oxidation potential, organic fertilizer could be an option to minimize or even upturn the negative impact of fertilization on methane oxidation of tropical soils, which would be a target of future research.



**Data availability**

The data generated in this study are available from the corresponding authors upon reasonable request.

**Supplement**

The supplement related to this article is available online.

**Author contributions**

JM: conceptualization (lead); investigation (lead); methodology (lead); supervision (lead); formal analysis (lead); writing (original draft) (lead) and writing (review and editing) (equal). KS: investigation (equal); methodology (equal); writing (review

and editing) (equal). CD: investigation (equal); methodology (equal); writing (review and editing) (equal). OD: investigation (supporting); writing (review and editing) (equal). RC: resources (equal); investigation (supporting); writing (review and editing) (equal). WR: investigation (supporting); writing (review and editing) (equal). WA: investigation (supporting); writing (review and editing) (equal). MS: investigation (supporting); writing (review and editing) (equal). PK: investigation (supporting); writing (review and editing) (equal). DE: conceptualization (supporting); investigation (equal); writing (review

and editing) (equal).

**Competing interests**

The contact author has declared that none of the authors has competing interests.

**Acknowledgments**

This research acknowledges the KAKENHI Grant-in-Aid for Fund for the Promotion of Joint International Research (Fostering Joint International Research(B)) to DE. This work was also financially supported by the Office of the Ministry of Higher Education, Science, Research and Innovation and the Thailand Science Research and Innovation through the Kasetsart University Reinventing University Program 2023. The authors are grateful to the staff of SKRS and CRRC for their help in

the field survey.

**Financial support**

This research has been supported by the KAKENHI Grant-in-Aid for Fund for the Promotion of Joint International Research (Fostering Joint International Research(B)) (grant no. 21KK0114).

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
