# Peer review of "Methane oxidation potential of soils in a rubber plantation in Thailand affected by fertilization"

_EGUsphere, 2024_

## Author Comment (AC2)

>General comments:

This manuscript presents data on soil potential methane oxidation rates (PMORs) from incubation experiments of rubber plantation soils in Thailand. PMORs are assessed using lab incubations of soil samples with enhanced methane concentrations. PMOR appears to be negatively correlated with fertilizer application rate. In-situ measurements of methane uptake also appear to negatively correlate with fertilizer application rate. Interestingly, PMORs from the top 10 cm of soil are found to be smaller than in-situ soil methane uptake. PMORs measured at several depths down to ~50 cm indicate that PMORs may be higher in the subsurface. These measurements also tend to show PMOR throughout the soil profile with the higher fertilizer application rates. PMORs are aggregated throughout the soil profile to produce a single per-area value.

This manuscript provides some insightful data on the oxidation of methane by soils in tropical rubber plantations. The finding of fertilizer suppressing methane oxidation in the study area is presented more or less convincingly. Additionally, the discussion of PMORs throughout the soil column potentially sheds light on the interplay of biological and physical processes leading to methane uptake by soil. The introduction is well-organized and relevant, providing good context for the study.

However, this manuscript has shortcomings in several areas. The experimental design and methods are lacking thorough description. The in-field treatments, the conditions of the incubations, and the steps for aggregating soil profiles are all rather opaque. The discussion does not adequately address complex and potentially interesting findings, as in the finding that total soil nitrogen is correlated with higher methane oxidation. The dataset could benefit from some editing, such as with the inclusion of some data from a palm plantation. The data visualizations are straightforward, but have room for improvement to concisely and meaningfully present the main findings. In terms of writing, the overall organization is good, but the manuscript would benefit from thorough copy editing to improve the use of English language for clarity and readability.

**Thank you so much for your positive and constructive feedback on our manuscript. We revised the manuscript based on your comments. Especially, we added more information on methods and clarified the difference between in-situ methane fluxes and estimated oxidation potentials and improved the discussion. We have carefully edited the final version of the revised manuscript to improve the quality of the English. We hope the revised manuscript is suitable for publication.**

**We reply to your valuable comments as follows. The MS word file is also attached for easier visibility.**

>Specific comments:

- The verb tenses vary between past and present throughout the manuscript. Please standardize here

→**We have carefully edited the final version of the revised manuscript to improve the quality of the English grammar.**

- Line 24: "potential methane oxidation potential"

→**corrected**

- Line 70-71: results should be removed from the discussion

→**We agree that the results should be removed from the INTRODUCTION, and we clarified the tested hypotheses instead.**

*"In this study, we measured the potential rates of soil methane oxidation using a microcosm incubation experiment with the hypothesis that land use change and fertilization management influence methane oxidation in tropical forest soil, focusing on a para rubber plantation. While most studies assume that methane oxidation in forest soils occurs primarily in the surface soil, we also targeted the deeper soil layers and tested the hypothesis that the influence of topdressing fertilizers on soil methane oxidation reaches deeper layers of the soil profile."*

- Line 81: The type of fertilizer and its N-P-K values should be reported, as well as its approximate application rate per hectare.

→**We revised the table accordingly.**

- Line 103-104: The preparation of MORP samples should be much expanded, using something like the following reference as an example for writing:
- Line 107-108: The amount of methane (0.5 or 1.0ml into 50ml or 100ml) doesn't add up with the reported 50ppm in the incubation experiments - it should be 100ppm, unless I'm missing something. Also, the use of such high methane concentration comes with some cause for concern and should at least be

discussed, since it is much higher than atmospheric mixing ratios.

→**Thank you very much for pointing our mistake. The correct volumes of methane are 0.25 or 0.5 ml. The method of PMOR measurement was revised.**

- Oxygen limitation is another potential issue with these incubations, but it seems promising that at least some of the incubations fully oxidized the methane.

    →**We incubated the sieved soils at the atmospheric level of oxygen, and the soil samples with high methane oxidation potentials showed a linear decrease in methane concentration from the beginning of the incubation period even though the soils contained higher organic carbon ranging from 10-23 g kg$^{-1}$. In addition, no correlation between PMOR and soil organic carbon content was observed (Supplementary Figure S2). Thus, we consider that oxygen was not a limiting factor for the measurements of methane oxidation potential in this study.**

- Line 113: "Adding up the methane oxidation rates" needs to be described mathematically to show what has been done. More broadly, I'm not sure this technique fully respects the actual field processes, ie the concentrations of methane and oxygen at depth, and the exchange of gas with the atmosphere.

    →**We fully agree with your argument. We consider the methane oxidation potential could be overestimated compared to the in-situ oxidation because of the high methane and oxygen concentrations in the incubation vials. We aimed to compare methane oxidation potentials by incubation experiment with the in-situ flux. We described in detail the method for calculating the potential methane oxidation rate per area:**

*"The potential methane oxidation rate per area (PMOR$_{area}$, nmol m$^{-2}$ s$^{-1}$) was estimated by summing the methane oxidation rates of different layers.*

$$PMOR_{area} = \sum_{l=1}^{n}(PMOR \times BD \times Th) \times 10{,}000 \div 3600 \div 16$$

*where l is the soil layer, PMOR (ngCH$_4$ g$^{-1}$ dry soil h$^{-1}$) is the potential methane oxidation rate, BD is the bulk density (1.5 g cm$^{-3}$), Th (cm) is the thickness of the soil layer, 10,000 converted cm$^{-2}$ to m$^{-2}$, 3600 converted h$^{-1}$ to s$^{-1}$, and 16 is the molar mass of methane."*

- Line 119: Methane production potential is mentioned in the discussion but not measured here, as in other works. This should also be at least discussed, as it potentially confounds some of the main findings.

  **→We agree with your arguments. The net methane flux from the soil is the balance between production and oxidation. A detailed study is needed, and this point is added in the conclusion of the revised manuscript.**

  *"In this study, we adapted the sampling strategy over time due to the fact that the topsoil has a low methane oxidation potential, unlike previous studies, and thus, we targeted the deeper layers in the middle of the study; a more systematic study is necessary for the future, where high-affinity methane oxidation and methane production should be addressed. The increase in methane oxidation with depth can be related to a shift in the composition of the methanotrophic community from high- to low-affinity methanotrophs, which remains to be studied. Nevertheless, our results provide a new insight into the impact of agricultural land use of tropical forests on the ecological function in a greenhouse gas cycle."*

- Line 120-124: The sample collection, preservation, preparation, and analysis all need to be better described. The sample state gives important context to the chemical analysis.

  **→We added more detailed information about sample collection, preservation, preparation, and analysis.**

  *"The soil samples were sieved (< 2 mm) on site and stored at room temperature to measure methane oxidation potential within a month. The sieved soil samples for chemical analysis were stored at 4°C."*

- Table 2 can and should be converted to a figure, as it represents the main findings of the manuscript

  **→We like to keep the table because 1) we need to compare the flux (minus data) and PMOR and 2) we think it is important to show the values.**

- Figure 4 is presented inconsistently - why are all treatments lumped together for February 2024, but Tr1 and Tr4 are separate panels in August?

  **→It is because we collected soil samples with depth with different intervals**

**between Tr1 and Tr4.**

---

## Author Comment (AC3)

**Thank you so much for your positive and constructive feedback on our manuscript. We revised the manuscript based on your comments. Especially, we added more information on the methods and clarified the difference between in-situ methane fluxes and estimated oxidation potentials. We hope that the revised manuscript is suitable for publication.**

**We reply to your valuable comments as follows. The MS word file is also attached for easier visibility.**

>Major concerns

The field fluxes measurements and microcosm experiment lack important information such as soil moisture and physical properties, which are highly relevant to methane processes in the soil. Considering the sandy-loam texture at the SKRS site, low methane oxidation potential in topsoil in the dry season might be due to low moisture and temperature. Did the authors measure and adjust moisture for soils from different sampling layers before incubation?

➔**The average air temperature at the SKRS site was 25.4, 27.2, and 26.5 °C in February 2023, August 2023, and February 2024, respectively. This was comparable with the temperature set for the incubation (25°C), and would thus not limit the methane oxidation potential. We added air temperatures in the materials and method. We measured the methane oxidation potential without adjusting the soil moisture. Soil moisture ranged from 5.1 to 12.7 % for the top layer (0-10 cm) and was not correlated with potential methane oxidation rates. We added the relationship between potential methane oxidation rate and soil water content in Supplementary Figure S2. The potential methane oxidation rate of the deeper layer soils was also not correlated with soil water content. We added a few sentences to the discussion on the potential inhibition of methanotrophic activities by water stress as described below. However, a low methane oxidation potential in the topsoil was also observed in the rainy season.**

*"Water balance is an important factor in regulating the methane dynamics in forest soils (Feng et al., 2020; Bras et al., 2022), but no correlation between the soil water content and PMOR was observed in this study. Either drought stress under low soil water content or limited oxygen under high soil water content can have an inhibitory*

*effect on soil methane oxidation (Feng et al., 2020), but the soil water contents measured in this study may have no such inhibitory effect."*

Bras, N., Plain, C., and Epron, D.: Potential soil methane oxidation in naturally regenerated oak-dominated temperate deciduous forest stands responds to soil water status regardless of their age—an intact core incubation study, Annals of Forest Science, 79, 10.1186/s13595-022-01145-9, 2022.

>It is worthy of adding more discussions on the gaps between in-situ methane fluxes and estimated PMORs, and possible reasons why the surface soil layer had lower PMORs than the subsoil layers, as well as how fertilization suppressed methane oxidation. Alternatively, adding an outlook after the conclusion about what needs to be done in future to address these questions.

→**Thank you for your helpful feedback.**

**The discrepancies between in-situ methane fluxes and estimated PMORs can be related to the fact that the vertical gradients in methane and oxygen concentration that exist in-situ in undisturbed soil profiles are not reproduced in the ex-situ incubation in which the soil from each layer is exposed to the same concentrations. PMORs measured on subsoil samples may, therefore, overestimate the actual oxidation occurring in situ deep in the soil profile. Furthermore, the incubations were carried out at a much higher methane concentration than expected in the soil profile, although an alternative source of methane may exist in deeper layers if methanogenesis occurs there. In this case, methanogenesis in the deeper layers may sustain methane oxidation in the upper soil layers by supplying the substrate (methane). We discussed these fundamental differences between the two approaches in the revised manuscript.**

"*The estimated rates ranged between 0.24 (Tr4) and 2.21 (Tr1) nmol $m^{-2}$ $s^{-1}$, which exceeded the in-situ fluxes. The gaps between in-situ methane fluxes and estimated PMORs can be related to the fact that the vertical gradients of methane and oxygen concentration that exist in situ in undisturbed soil profiles are not reproduced in the ex-situ incubation in which soil of each layer is exposed to the same concentrations. PMORs measured on subsoil samples may, therefore, overestimate the actual oxidation occurring in situ deep in the soil profile (Bender and Conrad, 1994). Another possible explanation is that the in-situ fluxes represent net methane uptake, i.e., the balance*

*between oxidation and production, thus could be lower than the oxidation rate.*"

Bender, M. and Conrad, R.: Methane oxidation activity in various soils and freshwater sediments: Occurrence, characteristics, vertical profiles, and distribution on grain size fractions, Journal of Geophysical Research-Atmospheres, 99, 16531-16540, 1994.

**The possible reasons why the surface soil layer had lower PMORs than the subsoil have been added to the discussion. Please see our answer to your specific comments about lines 161-163**

**We also added more discussion on how fertilization can suppress methane oxidation in the revised manuscript.**

*"Ammonium competitively suppresses methane monooxygenase due to the similarity with ammonia monooxygenase. Nitrate is also reported to strongly inhibit the atmospheric methane oxidation in forest soils (Mochizuki et al., 2012). Both ammonium and nitrate fertilizers are applied in the rubber plantation in this study, which likely suppressed methane oxidation. In addition to the high amount of fertilization, recurring and prolonged disturbances of methane oxidation by fertilization in Tr3 and Tr4 may outcompete the resilience of methane oxidation (Lim et al., 2024). Notably, fertilizers applied on the surface had a suppressive effect on methane oxidation in the deeper layers, at least up to 60 cm. Soil acidification is another possible cause of suppressed methane oxidation of forest soil by fertilization (Benstead and King 2001), but there is no relationship between soil pH and potential methane oxidation rate in this study."*

Benstead, J. and King, G. M.: The effect of soil acidification on atmospheric methane uptake by a Maine forest soil1, Fems Microbiol Ecol, 34, 207-212, 10.1111/j.1574-6941.2001.tb00771.x, 2001.

Lim, J., Wehmeyer, H., Heffner, T., Aeppli, M., Gu, W., Kim, P. J., Horn, M., and Ho, A.: Resilience of aerobic methanotrophs in soils; spotlight on the methane sink under agriculture, Fems Microbiol Ecol, 10.1093/femsec/fiae008, 2024.

Mochizuki, Y., Koba, K., and Yoh, M.: Strong inhibitory effect of nitrate on atmospheric methane oxidation in forest soils, Soil Biology and Biochemistry, 50, 164-166, https://doi.org/10.1016/j.soilbio.2012.03.013, 2012.

>Specific comments

>line 28: change off-season, use consistent terms for the seasons

**➔Changed to dry (off-harvesting) season**

>line 23: delete the first potential

**➔Thank you for your notice. The first one was deleted.**

>lines 28-30: Although the integrated potential numbers (Figure 5) might match with in-site measured methane fluxes, considering very different methane and oxygen concentrations in deeper soil layers under the field condition and incubation setting, be cautious to conclude that methane oxidation in the soil predominantly occurs in depths below the surface layer based on only one site. I suggest the authors present the integrated numbers as an additional column in Table 2. A clearer distinction between potential and in-situ rate should be made throughout the texts.

**➔We divided our study into two parts: 1) low methane oxidation potential in the topsoil layer under different land uses and fertilization levels for a para rubber plantation and 2) the depth profile of methane oxidation potential of para rubber soils influenced by fertilization. We want to keep the structure to highlight our two findings. We double-check the texts to ensure a clear distinction between potential oxidation rates and net in-situ fluxes.**

**We agree that we cannot conclude that soil methane oxidation predominantly occurs at depths below the surface layer based on ex-situ incubation alone because, as mentioned above, we are aware that PMORs measured on subsoil samples may overestimate the actual oxidation occurring in-situ deep in the soil profile.**

>lines 64-71: could you formulate them into hypotheses? Line 165 mentioned the hypothesis.

**➔We added two hypotheses in the last section of the introduction:**

*"In this study, we measured potential soil methane oxidation rates using a microcosm incubation experiment to test the hypothesis that land-use change and fertilization management influence soil methane oxidation in tropical tree plantations focusing on a para rubber plantation. While most studies assume that methane oxidation in forest soils occurs primarily in the surface soil, we also targeted the deeper soil layers and tested the hypothesis that the effects of topdressing fertilizers on soil methane oxidation*

*extend to deeper layers of the soil profile."*

>lines 75-78: the duration of wet season and dry season is unclear, please specify the start and end of each season.

**→We revised the sentence, clarifying the duration of the wet season and dry seasons.**

>lines 81-83: how long have been the fertilizer treatments set up in the rubber plantation at SKRS? What are the fertilizer forms especially N applied in the treatments? If the fertilization treatments have been carried out for a long time, a gradient of soil properties might be already established between treatments.

**→We added information on the setting up of the plantation and the fertilizers applied.**

*"The rubber plantation of the SKRS site was set up in 2007 and has been applied with four different levels of fertilizer treatment with randomized four replicate blocks (A–D) since 2014 at the beginning of tapping: Tr1, no; Tr2, low; Tr3, intermediate; Tr4, high (Table 1). Tr3 falls within the range of the recommended fertilizer application rates for mature rubber plantations in Thailand by Thai public institutions; recommendations exceeded by 40% of rubber farmers (Chambon et al., 2018). Chemical fertilizers of nitrogen (40% nitrate and 60% ammonium), phosphorus, and potassium (YaraMilaTM, Yara International ASA, Oslo, Norway) are top-dressed in the wet season, evenly to half of the area between the planting rows. The fertilizer was applied only in the early rainy season (May) for T2 while a second application was made late in the rainy season (October) for T3 and T4."*

>lines 95-99: it is interesting to see how field sampling progressively changed over time, at the same time, it limited what statistical analysis could test, e.g. seasonal effect, land use effect, interactions, etc.

**→Thank you for the comments. We first focused on the topsoil layer, assuming it should be the most active part of methane oxidation, as reported in different**

**studies. However, our initial experiment in February 2023 indicated that the surface soils had only limited methane oxidation potential compared to in-situ methane fluxes. We, therefore, adapted the sampling strategy over time. Since samples collected from deeper soil layers in August 2023 showed higher potential methane oxidation, we intensified this sampling along the vertical profile in February 2024, our last field campaign in this overseas project. Despite some limitations, we believe that we report new findings in this short paper, but we added a sentence on the necessity for a systematic study in the conclusion of the revised manuscript.**

*"In this study, we adapted the sampling strategy over time due to the fact that the topsoil has a low methane oxidation potential, unlike previous studies, and thus, we targeted the deeper layers in the middle of the study; a more systematic study is necessary for the future, where high-affinity methane oxidation and methane production should be addressed. The increase in methane oxidation with depth can be related to a shift in the composition of the methanotrophic community from high- to low-affinity methanotrophs, which remains to be studied. Nevertheless, our results provide a new insight into the impact of agricultural land use of tropical forests on the ecological function in a greenhouse gas cycle."*

>lines 106-107: sieved fresh soil? Which samples were put into 50-ml GC vials? Considering the long incubation time (30 days) in this study, was it possible oxygen became limited during the incubation? The limitation of using high initial methane concentration in incubation should be communicated to readers, i.e. not favoring high-affinity methanotrophs that oxidize low concentrations of methane (more dominant in aerated soils). This might be one of the reasons for the low estimation of oxidation potential.

**→We incubated the sieved soils at the atmospheric level of oxygen, and the soil samples with high methane oxidation potentials showed a linear decrease in methane concentration from the beginning of the incubation period even though the soils contained higher organic carbon ranging from 10-23 g kg$^{-1}$. Furthermore, no correlation between PMOR and soil organic carbon content was observed (Supplementary Figure S2). Thus, we consider that oxygen was not a limiting factor for the measurements of methane oxidation potential in this study.**

**It is true that high-affinity methanotrophs can be saturated at much lower methane concentration than 50 ppm, but to our knowledge, there is no report that**

**high methane concentration would inhibit their oxidation capacity. However, we agree that the increase in methane oxidation with depth can be related to a shift in the composition of the methanotrophic community from high- to low-affinity methanotrophs. The fact that high-affinity methanotrophs were found to be less sensitive to nitrate than low-affinity methanotrophs (Reay et al., 2005) is consistent with our observation that the effect of fertilization was more evident in the subsoil than in the topsoil. However, another study reported a negative relationship between mineral nitrogen and methane oxidation at ambient concentration and a positive relationship at elevated methane concentration (Chan and Parkin, 2001). The composition of the methanotrophic community deserves to be addressed in future research.**

**We added this suggestion for future research in the conclusion of the revised manuscript as you may see above.**

>lines 152-155: higher total N correspond to higher PMORs? This seems contradictory to the negative fertilization effect on PMORs and in-site methane fluxes (figure 2, lines 142-144). Could the authors add the surface soil (0-10 cm) properties by treatment to Table 2 or in supplement? I do not understand the argument here either, is organic fertilizer applied in this study?

**→A supplementary table will be given for soil properties. No organic fertilizer was applied; we clarified this in the revised text.**

>Figure 2: what does 'corrected' mean?

**→Sorry for the typo, it is "collected."**

>line 157: medium is more suitable than middle

**→Corrected**

>line 161-163: very important observation, it is worth discussing possible reasons for the gap between PMOR and methane flux in situ.

**→We assumed that the PMOR would be higher than in situ soil methane uptake, because the methane concentration used in incubation vials was much higher than the atmospheric level. Nevertheless, the PMOR in the surface soils (0-10 cm) was low compared to the in-situ methane flux. Although negative artifacts on methane oxidation in the incubation experiment cannot be completely ruled out, the results**

**directed us to focus on soils from the deeper layers, which was addressed in the following section. We added the following sentences:**

*"The estimated aerial PMORs of the surface soil (0-10 cm) were much lower than the methane fluxes measured on site in February 2023 during the dry season; the same trend was observed in Tr1 and Tr2 in August 2023 during the wet season. PMORs measured in this study likely overestimate the actual oxidation because the initial methane concentration (50 ppmv), higher than the atmospheric level, would accelerate methane oxidation (Bender and Conrad 1994). Thus, the significant discrepancy between the topsoil PMOR and the in-situ methane uptake suggests that the methane oxidation in the topsoil does not explain the in-situ methane uptake in the studied para rubber plantation."*

**The discrepancy between the in situ methane flux and PMOR per area including the deeper soil layers, i.e., higher PMOR per area than the in-situ flux, has also been discussed in the following section as you may see above.**

>Figure 3: I think keeping one set of legends is sufficient here because of the same sampling depths.

**➔Done**

>lines 196-199: what are the bases for this statement? The correlation in Figure S2 was total N and the authors did not mention organic fertilizers in the methods description at all.

**➔No organic fertilizers were added to the plot. We meant the soil organic matter. We clarified it in the text.**

---

## Author Response (AR1)

Manuscript number: egusphere-2024-293730 Sep 2024

Title: Methane oxidation potential of soils in a rubber plantation in Thailand affected by fertilization

Response to Topic Editor

Dear Dr. Emily Solly,

Thank you very much for editing our manuscript. We appreciate your helpful comments after the manuscript was reviewed. The manuscript was revised according to the comments from you and the reviewers. Please see the changes in red in the author's track-changes file.

The referees found that your manuscript presents novel information about the oxidation of methane of soils in tropical rubber plantations, in particular providing knowledge on the overlooked subsoils and the responses to changes in fertilization regime. However, they also raised several points that still should be addressed. In particular, both referees had some concerns regarding the lack of provision of some aspects of your sampling design, the reporting of the environmental conditions during the microcosm study, and the aggregation of the soil samples. This information is highly relevant to explain methane processes in the soil.

We addressed all the points the reviewers raised and revised our manuscript. We would appreciate it if you kindly check our reply on the reviewers' comments (AC2 and AC3) in the interactive discussion, where we also clarified how we revised our manuscript.

In additional to several other specific comments, the referees also found that the discussion is currently lacking to report some potentially interesting findings, such as:
1) the observation the soil nitrogen was strongly correlated with higher methane oxidation,

We described the observation in the revised manuscript as follows:

*"A weak positive correlation with total nitrogen was detected, and principal component*

*analysis demonstrated that the PMOR and total nitrogen had similar eigenvalues, driven by Tr1 soils, particularly in the wet season (August 2023, Supplementary Figure S3). The results may imply that soil organic nitrogen slowly supplies inorganic nitrogen at a rate that does not suppress but supports methane-oxidizing bacteria (Geng et al., 2017) in contrast to the high application of chemical fertilizers that often suppress methane oxidation   (Liu and Greaver, 2009; Aronson and Helliker, 2010)."* (L 168-173)

**2) the differences between in-situ methane fluxes and estimated PMORs,**

We described the differences as below:

*"The estimated aerial PMORs of the surface soil (0-10 cm) were much lower than the methane flux on site in the dry season; the same trend was observed in Tr1 and Tr2 in the wet season. PMORs measured in this study likely overestimate the actual oxidation as the initial methane concentration (50 ppmv) higher than the atmospheric level would accelerate methane oxidation (Bender and Conrad, 1994). Thus, the significant gap between the PMOR of topsoil and the methane uptake in situ suggests that the methane oxidation in the topsoil does not explain the in-situ methane uptake in the para rubber plantation studied."* (L. 184-188)

*"The estimated rates ranged between 0.24 (Tr4) and 2.21 (Tr1) nmol m-2 s-1, which exceeded the in-situ fluxes. The gaps between in-situ methane fluxes and estimated PMORs per area can be related to the fact that the vertical gradients of methane and oxygen concentration that exist in situ in undisturbed soil profiles were not reproduced in the ex-situ incubation in which soil of each layer was exposed to the same concentrations. PMORs measured on subsoil samples may, therefore, overestimate the actual oxidation occurring in situ deep in the soil profile (Bender and Conrad, 1994). Another possible explanation is that the in-situ fluxes represent net methane uptake, i.e., the balance between oxidation and production, thus could be lower than the gross oxidation rate."* (L. 221-227)

**3) possible reasons why the surface soil layer had lower PMORs than the subsoil layers**

Although further investigation is of course needed, we discussed some potential reasons as follows:

*"Our findings contrast with previous studies that reported higher high- and low-affinity methane oxidation in the topsoil than below, though some exceptions were noticed*

*when high mineral N concentrations were measured in the topsoil (Reay et al., 2005; Xu et al., 2008). However, in our study, the discrepancy between in-situ soil methane uptake and PMOR was observed in all treatments, including T1, although the gap was less pronounced in T1 than in the three other treatments receiving fertilization. Low soil water content, especially during the dry season, can be another factor pushing methane oxidation down to the soil profile since drought stress is known to inhibit methanotrophic activity (Schnell and King, 1996; Borken et al., 2006; Bras et al., 2022). However, the discrepancy between in situ soil methane uptake and PMOR was observed in all seasons at our site. Alternatively, methane oxidation can be inhibited by several chemical compounds such as monoterpenes and ethylene that can be abundant in the upper soil layer under several types of vegetation (Amaral and Knowles, 1998; Jäckel et al., 2004; Maurer et al., 2008). While we did not assess the presence of potential inhibitors of methane oxidation in our study, this hypothesis cannot be ruled out.*" (L. 228-238)

**4) the reason why fertilization suppressed methane oxidation**
We discuss the possible reasons as below:

"*Fertilization, especially nitrogen fertilizer application, is often reported to inhibit soil methane oxidation (Täumer et al., 2021; Bodelier, 2011; Bodelier and Laanbroek, 2004). Ammonium competitively suppresses methane monooxygenase due to the similarity with ammonia monooxygenase. Nitrate is also reported to strongly inhibit atmospheric methane oxidation in forest soils (Mochizuki et al., 2012). Both ammonium and nitrate fertilizers are applied in the rubber plantation in this study, which likely suppressed methane oxidation. In addition to the high amount of fertilization, recurring and prolonged disturbances of methane oxidation by fertilization in Tr3 and Tr4 may outcompete the resilience of methane oxidation (Lim et al., 2024). Notably, fertilizers applied on the surface had a suppressive effect on methane oxidation in the deeper layers, at least up to 60 cm. Soil acidification is another possible cause of suppressed methane oxidation of forest soil by fertilization (Benstead and King, 2001), but there is no relationship between soil pH and potential methane oxidation rate in this study.*" (L. 212-220)

**5) an indication of future research directions**

We discuss it in the conclusion section:

"*A more systematic study is necessary for the future, where high-affinity methane oxidation and methane production should also be addressed. The increase in methane oxidation with depth can be related to a shift in the composition of the methanotrophic community from high- to low-affinity methanotrophs, which remains to be investigated.*" (L. 241-244)

"*As soil organic nitrogen weakly but positively correlated with soil methane oxidation potential, soil enrichment with organic nitrogen, e.g., by organic fertilizer application, may be an option to minimize or even reverse the negative impact of fertilization on methane oxidation of tropical soils, which should be a target of future research.*" (L. 248-251)

---

## Author Response (AR3)

Before the manuscript can be formally accepted, I have some additional remarks which should be addressed. In particular, it is crucial that the data associated to your manuscript is made openly available.

We are grateful for your careful editing of our manuscript. We accordingly revised to your comments.

1) To improve the readability of the results referring to the SKRS site (especially of the figures and tables), it would help to highlight that the Tr1-4 refer to different fertilization treatments in all relevant figure legends.
-->We added the phrase "under different fertilizer treatments" for the figures and table.

Also, in the methods section you could write: 'The rubber plantation of the SKRS site was set up in 2007, and four different levels of fertilization have been applied on randomized four replicate blocks (A–D) since 2014 at the beginning of tapping. The four fertilization treatments (Tr) refer to: Tr1, no; Tr2, low; Tr3, intermediate; Tr4, high fertilizer application rates (Table 1)'.
-->Corrected as suggested.

In Table 1, it would be helpful to rename the column 'Treatment' to 'Fertilization treatment'
-->Corrected as suggested.

2) The Latin name of Para rubber should be given correctly as (Hevea brasiliensis Müll. Arg.) Hevea brasiliensis kept in italic, as you provided.
-->Corrected as suggested.

3)The beginning of conclusion still requires some improvement, as the first three sentences are currently an abrupt shift from the discussion. Suggestion: Start the conclusion by highlight the main findings of the study, and afterwards mention how the adopted sampling strategy indicated the relevance of investigating these studied processes not only in topsoils but also at deeper soil depths, etc...
--> We accordingly revised the beginning of conclusion as follows:
"The present study demonstrates that the high fertilization negatively impacts the methane oxidation potential of soils in the Para rubber plantation. The top dressing of fertilizer suppressed methane oxidation not only in the topsoils but also in the soils with deeper layers, which may significantly contribute to the methane cycle in the soil column. In this study, we adapted the sampling strategy over time since the topsoil

we collected in our first sampling showed an unexpectedly low methane oxidation potential, unlike previous studies."

4) Last but not least, 'Copernicus Publications requests depositing data that correspond to journal articles in reliable (public) data repositories, assigning digital object identifiers, and properly citing data sets as individual contributions'. More information can be found here: https://www.soil-journal.net/policies/data_policy.html

Therefore, it is not enough to simply stating that 'The data generated in this study are available from the corresponding authors upon reasonable request'. It is necessary to make the data associated to your manuscript openly available in a reliable public data repository, or as supplementary data table(s).

--> We submitted the data of the figures as an Excel file.

Once you have made the required revisions and made the data openly available, please proceed to upload a revised version of the manuscript

Kind regards,
Emily Solly

Authors' response to suggestions from the editor

-please adapt the color/symbol coding in the figures of the main text and supplementary to ensure consistency across the same treatments (i.e. the color/symbol of each treatment should be the same in all figures)

-->We tried our best to keep the consistent use of the color/symbol.

-when results related to soil depth are shown (e.g. in Fig. 3, Fig. S4, Fig. S5) select different colors than those you used for the treatments (or alternatively just use a symbol-based legend) --

-->We used a symbol-based legend to show results related to soil depth and seasonality: Fig. 3, Fig. S3, Fig. S4, and Fig. S5.

-check that the text/legend/axes style you used is the same in all figures

-->We checked the text/legend/axes style and corrected keep the consistent use.

-please remove the grid lines in Fig. 4 and Fig S6, as these are not present in the other figures ---

-->done

-in the Legend of Fig 3., remove the wording 'Tr4'

-->Thank you. We removed them.

Authors' response to suggestions from the reviewer #1

Thank you to the authors for your revisions to the manuscript, which significantly improved the clarity of presentation, quality of discussion, and precision of the methods section. The main findings are presented more clearly and with better context, and I suggest only minor revisions to the presentation and discussion.

-->Thank you so much for your positive feedback on our revised manuscript.

Since the hypothesis of deeper methane oxidation is shown as a primary result, it would appropriate to systematically report depth-integrated PMOR-area as calculated with the equation at line 120, perhaps as a third column to Table 2.

-->The depth-integrated PMOR-area is indeed a highlight of this study, and thus, we want to show the result in a separate figure (Figure 5) to effectively demonstrate the effect of fertilizer application.

There is still room for improvement in the discussion of the interesting finding of MORP increasing with depth. It is not clear from the discussion how this adds to the current knowledge of depth-resolved MORP, aside from brief discussion on lines 185-186 and 221-227. Discussion of what the actual Oxygen and CH4 concentrations in the soil profile may be is crucial for interpreting the results of lab incubations. This is especially of note given that some treatments are observed producing methane during during some sampling times.

-->Thank you for your suggestion. We added the following sentences including the information on methane concentration in the soil at the study site, which is given in our latest submission:

"*Methane oxidation in the soil is insensitive to a wide range of oxygen levels (2-20%) but suppressed at extremely low oxygen levels (<2 %) (Walkiewicz and Brzezińska, 2019; Walkiewicz et al., 2018; Bender and Conrad, 1994). Methane concentration in the soil of the study site is often lower than the atmospheric level and much less compared to that in the ex-situ incubation, even in the hotspots of methane accumulation during the wet season (3.76 ppm) (Epron et al., 2025). "* (L 22-228)

Epron, D., Chotiphan, R., Wang, Z., Duangngam, O., Shibata, M., Paul, S. K., Mochidome, T., Sathornkich, J., Azuma, W. A., Murase, J., Nouvellon, Y., Kasemsap, P., and Sajjaphan, K.: Fertilization turns a rubber plantation from sink to methane source, EGUsphere, 2025, 1-35, 10.5194/egusphere-2025-

2, 2025.

Walkiewicz, A. and Brzezińska, M.: Interactive effects of nitrate and oxygen on methane oxidation in three different soils, Soil Biol Biochem, 133, 116-118, https://doi.org/10.1016/j.soilbio.2019.03.001, 2019.

Walkiewicz, A., Brzezińska, M., and Bieganowski, A.: Methanotrophs are favored under hypoxia in ammonium-fertilized soils, Biol Fert Soils, 54, 861-870, 10.1007/s00374-018-1302-9, 2018.

Additionally, the manuscript would still benefit from thorough proofreading to check grammar and language. Particularly, "para rubber" should be changed to "Pará rubber" throughout, and the latin name H. Brasiliensis should be mentioned at least once. Also, "in-situ" and "ex-situ" should be spelled consistently either with or without a hyphen.

-->Thank you for your suggestions. We change "para rubber" to "Para rubber" throughout the manuscript. We also added the latin name of Para rubber in the section of the study site (L72). We used "in-situ" as a compound adjective and "in situ" as an adverb according to a grammatical instruction. We check the consistency of the use throughout. We also made additional proofreading to check grammar and language.